# Structural Diversity among *Edwardsiellaceae* Core Oligosaccharides

**DOI:** 10.3390/ijms24054768

**Published:** 2023-03-01

**Authors:** Maria Jordán, Sylwia Wojtys-Tekiel, Susana Merino, Juan M. Tomás, Marta Kaszowska

**Affiliations:** 1Department of Genetic, Microbiology and Statistic, University of Barcelona, Diagonal 643, 08028 Barcelona, Spain; 2Laboratory of Microbial Immunochemistry and Vaccines, Ludwik Hirszfeld Institute of Immunology and Experimental Therapy, Polish Academy of Sciences, 53-114 Wroclaw, Poland

**Keywords:** *Edwardsiellaea*, genomic, core oligosaccharide, NMR spectroscopy

## Abstract

The *Edwardsiella* genus presents five different pathogenic species: *Edwardsiella tarda*, *E. anguillarum*, *E. piscicida*, *E. hoshinae* and *E. ictaluri*. These species cause infections mainly in fish, but they can also infect reptiles, birds or humans. Lipopolysaccharide (endotoxin) plays an important role in the pathogenesis of these bacteria. For the first time, the chemical structure and genomics of the lipopolysaccharide (LPS) core oligosaccharides of *E. piscicida*, *E. anguillarum*, *E. hoshinae* and *E. ictaluri* were studied. The complete gene assignments for all core biosynthesis gene functions were acquired. The structure of core oligosaccharides was investigated by ¹H and ^13^C nuclear magnetic resonance (NMR) spectroscopy. The structures of *E. piscicida* and *E. anguillarum* core oligosaccharides show the presence of →3,4)-L-*glycero*-α-D-*manno*-Hep*p*, two terminal β-D-Glc*p*, →2,3,7)-L-*glycero*-α-D-*manno*-Hep*p*, →7)-L-*glycero*-α-D-*manno*-Hep*p*, terminal α-D-Glc*p*N, two →4)-α-D-Gal*p*A, → 3)-α-D-Glc*p*NAc, terminal β-D-Gal*p* and →5-substituted Kdo. *E. hoshinare* core oligosaccharide shows only one terminal β-D-Glc*p*, and instead of terminal β-D-Gal*p* a terminal α-D-Glc*p*NAc. *E. ictaluri* core oligosaccharide shows only one terminal β-D-Glc*p*, one →4)-α-D-Gal*p*A and do not have terminal α-D-Glc*p*N (see complementary figure).

## 1. Introduction

One of the less frequently encountered pathogenic genera in the order *Enterobacteriales* is the genus *Edwardsiella* [1], which was established in 1965 by Ewing et al. [2]. It belongs to the order and family *Hafniaceae* based upon biochemical and physiologic characteristics, the presence of the enterobacterial common antigen, and different molecular signatures identified through comprehensive comparative genomic analyses [3,4]. *E. tarda*, *E. ictaluri* and *E. hoshinae* have been the traditional causative agents of Edwardsiellosis until 2012; however, intensive studies have just recently revealed two other species, *E. piscicida* and *E. anguillarum* [5].

Edwardsiellosis is a serious disease affecting a wide range of cultured fish species both in marine and freshwater environments [6]. It usually occurs under imbalanced environmental conditions, such as high water temperatures, poor water quality and high organic content. Fish infected show abnormal swimming behavior, loss of pigmentation, swelling of the abdominal surface, rectal hernia and other symptoms [7]. This disease may become a significant health issue for fish and humans, accounting for severe economic losses in the aquaculture industries [8].

*E. tarda* is the best studied species in this genus, having a wide ecological niche and host range including fish, birds, amphibians, reptiles, marine mammals and humans [5]. In humans, is the only recognized pathogenic species primarily associated with sporadic cases of gastroenteritis. In rare instances, has also been reported to cause extraintestinal diseases, involving cases of septicaemia and bacteremia [1]. *E. ictaluri* causes enteric septicaemia in cultured channel and white catfish, which appear to be the primary host species of this pathogen, as well as tilapia and other warm-water species [9]. *E. hoshinae* has been most often isolated from reptiles and birds [10]. Two new species have been recently added to the *Edwardsiella* genus, first *E. piscicida* and later *E. anguillarum*. *E. piscicida* comprises exclusively pathogenic strains isolated from fish but shares many phenotypic characteristics identical to *E. tarda*. *E. anguillarum* is a microorganism potentially pathogenic to eels and distinguishable from the other species of the genus for the capacity to produce acetoin from glucose and to ferment arabinose [5].

The LPS is an amphiphilic molecule located in the outer leaflet of the outer membrane of gram-negative bacteria that confers stability, integrity and organization to the outer membrane. Furthermore, it maintains the barrier function against bacteriophages and the action of certain antibiotics, as well as against the host defense mechanism during infections [11]. The LPS molecule consists of a polysaccharide and a lipid A part. The lipid A is the hydrophobic part and has endotoxic and pyrogenic properties, among others. The polysaccharide is the hydrophilic part and can be subdivided into the O-specific chain (O-antigen), which is more external and variable, and the core oligosaccharide, which is internal and conserved [11]. The core oligosaccharide can be further subdivided into the inner core and outer core. The inner core is bound to the lipid A and commonly contains two or three heptoses and 2-keto-3-deoxyoctulosonic acid (Kdo), and in many bacteria, it contains non-carbohydrate components, such as phosphate, amino acids and ethanolamine substituents. The outer core usually contains hexoses or hexosamines and is the part bound to the O-antigen. Unlike the inner core, which tends to be evolutionary conserved within a taxonomic family or genus, the outer core shows more variability [12]. The O-antigen, which may be present (Smooth-LPS) or not (Rough-LPS), is the external component of the LPS and consists of repeating oligosaccharide units that may be linear or branched. It shows the largest variation between species and evokes a specific response [13,14].

The genes for the core oligosaccharide of LPS are mainly organized into clusters of contiguous genes. In several *Enterobacteriaceae,* such as *Escherichia coli*, *Salmonella enterica* and *Klebsiella pneumoniae*, genes involved in LPS core biosynthesis are found clustered in a region of the chromosome, the *waa* gene cluster [15]. However, in other species such as *Proteus mirabilis*, *Yersinia enterocolitica* and *Plesiomonas shigelloides*, belonging to different families of *Enterobacteriales*, some genes involved in LPS core biosynthesis are not clustered and located outside the *waa* gene cluster [16,17,18].

As a first, the genome of *E. tarda* EIB202 was fully sequenced, and a clear region with the *waa* gene cluster was identified. As in the majority of *Enterobacteriales,* the first gene of the cluster is *hldD* (also referred to as *rfaD*), which codes for the ADP-L-*glycero*-D-*manno*-heptose-6-epimerase (encoded protein ETAE_0083), and at the 3′ end of the cluster, the *coaD* (encoded protein ETAE_0071) codifies for phosphopantetheine adenylyltransferase. The function of genes found in this cluster seems to be in agreement with the chemical structure of the core oligosaccharide of *E. tarda* EIB202: →5-substituted Kdo*p*, →3,4)-L-*glycero*-α-D-*manno*-Hep*p*, →2,3,7)-L-*glycero*-α-D-*manno*-Hep*p*, two terminal β-D-Glc*p*, →7)-L-*glycero*-α-D-*manno*-Hep*p*, two →4)-α-D-Gal*p*A, terminal α-D-Glc*p*N, →3)-α-D-Glc*p*NAc and terminal β-D-Gal*p,* which had been previously described [19]. However, the LPS motif β-D-Glc*p*-(1→2)-α-L-Hep*p*II is not encoded by any of the glycosyltransferases found in the *waa* cluster. This motif is highly similar to the WapG of *P. shigelloides* and is encoded by a gene outside this cluster, *ETAE_RS09105* [19].

The complete structural analysis of the core oligosaccharides is of high importance for a better understanding of LPS biological activity and is a prerequisite for strategies aimed at the treatment of endotoxicosis. As the *Edwardsiella* genus has four other species reported, we characterized the chemical structure of the core oligosaccharide of *E. piscicida*, *E. anguillarum*, *E. hoshinae* and *E. ictaluri*, supported by genomic analysis.

## 2. Results

### 2.1. Comparative Genomics of Edwardsiella Strains

In *E. tarda,* as in many *Enterobacteriales,* the *waa* region starts from *hldD* (also referred to as *rfaD*) to *coaD* gene. Analysis of seven *E. tarda*, thirteen *E. piscicida*, two *E. anguillarum*, six *E. ictaluri* and two *E. hoshinae* strains with complete genome sequences available on the NCBI website, using the *E. tarda* EIB202 *coaD* and *hldD*, allowed us to locate the *waa* gene cluster in each genome. Comparative genomics of this region using Mauve software version 20150226 show that they are highly conserved in all species, with some differences in *E. ictaluri* and *E. hoshinae*, as well as in some *E. tarda* strains (Figure 1).

A phylogenetic tree generated by the neighbor-joining method on the basis of the *waa* cluster *(hldD* to *coaD*) sequence shows two different types of *waa* clusters in *Edwardsiella* (Figure 2). One is found in all *E. piscicida*, *E. anguillarum* and *E. ictaluri* strains tested, and in three *E. tarda* strains, including the EIB202 strain. The other type is found in all *E. hoshinae* tested and in four *E. tarda* strains.

To identify orthologs common in both species, reciprocal BLASTp (Figure 3) compared the complete set of predicted proteins of *E. tarda* EIB202 with that of the other. Although the proteins encoded by some genes in the *waa* cluster of *E. piscicida* and *E. anguillarum* have been annotated with different names, all of them show identical or similar protein size (nº of amino acids) and identities higher than 93.0% of those of orthologous genes in the *E. tarda* EIB202 *waa* cluster (Figure 3). These data suggest that *E. piscicida* and *E. anguillarum* probably have the same core-oligosaccharide structure as *E. tarda* EIB202.

The *E. ictaluri waa* cluster does not contain either the *wapC* or part of the *wapB.* The remaining *wapB* encodes a peptide of 129 amino acids that shows 96.6% identity with the carboxy-terminal end of WapB. Furthermore, the WaaL of *E. ictaluri* is smaller (218 amino acids) than the WaaL of *E. tarda, E. piscicida* and *E. anguillarum* (377 amino acids), but shows an identity of 91.7% compared to *E. tarda* EIB202.

For *E. hoshinae*, which also presents differences in this genomic region, it can be seen that it lacks the *wabK* gene, and the downstream of *waaC* presents a new gene, which encodes a glycosyltransferase belonging to the glycosyltransferase family 2. This new glycosyltransferase has no homology to WabK, so they are different proteins. They show 84.0% identity to the hyaluronan synthase (HyaD) of *E. tarda* NCTC13561, which is present in the *waa* cluster of this strain, and also 40.5% identity with WapD of *P. shigelloides*. These homologies suggest that the function of this new glycosyltransferase (named WahX) could be *N*-acetylhexosamine. Curiously, glycosyltransferases orthologous to WahX were found in some *E. tarda* strains such as KC-Pc-HB1, AT98-87 and FL95-01 (85.0, 70.9 and 85.1% identities, respectively) that do not have orthologous to WabK. Furthermore, as in *E. ictaluri*, *waaL* encodes a smaller protein (265 amino acids) than the WaaL of *E. tarda* EIB202*, E. piscicida* and *E. anguillarum*. This protein does not show homology with the WaaL of any *E. piscicida* or *E. anguillarum* strain, nor to *E. tarda* EIB202. However, it shows 75.5 to 78.5% homology to the WaaL of *E. tarda* strains containing the *wahX* in the *waa* cluster. These data suggest that *E. ictaluri* has a different and smaller core oligosaccharide structure than the other four *Edwardsiella* species. The data also suggest that the *E. hoshinae* core-oligosaccharide structure is different than that of *E. tarda* EIB202, *E. piscicida* and *E. anguillarum* but similar to *E. tarda* strains that lack the *wabK* gene (Figure 2).

Finally, it is known that the *E. tarda* EIB202 LPS motif β-D-Glc-(1→2)-α-L-HepII, corresponding to the WapG (ETAE_RS09105), is not encoded by any of the glycosyltransferases present in the *waa* cluster. To find orthologous proteins to this one, we analyzed the complete genomes of *E. piscicida*, *E. anguillarum*, *E. ictaluri* and *E. hoshinae* strains using BLASTp. Comparative genomic analyses using Mauve show that the genomic region containing this gene is highly variable in the *Edwardsiella* species (Figure 4). Orthologous to this protein were found in all *E. piscicida* strains, in the *E. anguillarum* strain C-5-1 and in *E. tarda* strains, whose *waa* cluster contains the *wabK* gene. However, neither strain of *E. ictaluri*, *E. hoshinae* nor *E. tarda* strains containing the *wahX* gene possess alleles orthologous to WapG. In order to confirm it, five contiguous genes above and under the *wapG* of *E. tarda* EIB202 were checked in these *Edwardsiella* species, and the results were either low query coverage and a high E-value or no significant similarity found.

### 2.2. Structural Analysis of Lipopolysaccharide of Edwardsiella Species

The LPSs of four *Edwardsiella* species were visually characterized by electrophoresis-gel tricine SDS-PAGE. The analyses showed that while *E. tarda* EIB202, *E. piscicida* HL9.1 and *E. anguillarum* 205/03 LPSs are smooth (S-LPS), *E. ictaluri* AL-15-01-CATFISH and *E. hoshinae* DSMZ 13771^T^ LPSs are smooth/rough (SR-LPS). However, *E. ictaluri* appears to have a significantly lower amount and number of repeating units than others (Figure 5).

The yields of LPSs from *E. piscicida* HL9.1, *E. anguillarum* 205/03, *E. hoshinae* DSMZ 13771^T^ and *E. ictaluri* AL-15-01-CATFISH bacterial masses were 0.5%, 1.0%, 0.9% and 1.1%, respectively. The mild acid hydrolysis of the *E. piscicida* and *E. ictaluri* LPSs yielded seven fractions, and the other *E. anguillarum* and *E. hoshinae* LPSs yielded eight fractions. The fractions consisting of unsubstituted core oligosaccharide (OS) were identified: in *E. piscicida* and *E. hoshinae* as fraction VI (OSVI), in *E. anguillarum* and *E. ictaluri* as fraction VII (OSVII). All investigations were carried out on OSVI and OSVII fractions isolated from *Edwardsiella* LPSs.

The comparison of anomeric regions of *Edwardsiella* cores (Figure 6) has shown different numbers of sugar residues.

#### 2.2.1. Structural Analysis of *E. piscicida* HL9.1 and *E. anguillarum* 205/03 Core Oligosaccharides

The *E. piscicida* OSVI had the same structure as the *E. anguillarum* OSVII; thus, the data for OSVI is not presented herein to avoid unnecessary duplication. The ^1^H−^13^C HSQC-DEPT spectrum of *E. anguilarum* OSVII contained main signals for eleven residues belonging to the core-oligosaccharide structure. The structural analysis showed the presence of →3,4)-L-*glycero*-α-D-*manno*-Hep*p* (residue **B**), two terminal β-D-Glc*p* (residues **C** and **H**), →2,3,7)-L-*glycero*-α-D-*manno*-Hep*p* (residue **D**), →7)-L-*glycero*-α-D-*manno*-Hep*p* (residue **E**), terminal α-D-Glc*p*N (residue **G**), two →4)-α-D-Gal*p*A (residues **F** and **I**), → 3)-α-D-Glc*p*NAc (residue **K**), terminal β-D-Gal*p* (residue **M**) and →5-substituted Kdo*p* (residue **A**) (Figure 6, Table 1).

The ^1^H−^1^H NOESY spectrum showed strong inter-residue cross-peaks between the transglycosidic protons: H−1 of **B**/H−5 of **A**, H−1 of **D**/H−3 of **B**, H−1 of **C**/H−4 of **B**, H−1 of **E**/H−7 of **D**, H−1 of **H**/H−2 of **D**, H−1 of **F**/H−7 of **E**, H−1 of **G**/H−4 of **F**, H−1 of **I**/H−3 of **D**, H−1 of **J**/H−4 of **I,** H−1 of **K/**H−4 of **I** and H−1 of **M**/H−3 of **K**. The presence of heterogeneity in OSVII was due to the partial presence of a structure with terminal α-D-Glc*p*N (residue **J**) instead of β-D-Gal*p*-(1→3)-α-D-Glc*p*NAc-(1→ fragment (residues **M** and **K**). These analyses allowed for the establishment of the same structure of *E. piscicida* and *E. anguillarum* core oligosaccharides as were identified in *E. tarda* EIB202 [19].

#### 2.2.2. Structural Analysis of *E. hoshinae* DSMZ 13771^T^ Core Oligosaccharide

The ^1^H−^13^C HSQC-DEPT spectrum of the *E. hoshinae* OSVI contained main signals for ten anomeric protons and carbons and Kdo spin systems, respectively. The analysis of *E. hoshinae* OSVI spectra showed the presence of →3,4)-L-*glycero*-α-D-*manno*-Hep*p* (residue **B**), β-D-Glc*p* (residue **C**), →3,7)-L-*glycero*-α-D-*manno*-Hep*p* (residue **D**), →7)-L-*glycero*-α-D-*manno*-Hep*p* (residue **E**), two →4)-α-D-Gal*p*A (residues **F** and **I**), two terminal α-D-Glc*p*N (residues G and **J**), →4)-α-D-Glc*p*NAc (residue **K**), terminal α-D-Glc*p*NAc (residue **L**) and →5-substituted Kdo*p* (residue **A**) (Figure 6 and Figure 7, Table 2).

Residue **A** was identified as the 5-substituted Kdo on the basis of characteristic deoxy proton signals at δ_H_ 1.93 ppm (H−3*ax*) and δ_H_ 2.26 ppm (H−3*eq*), as well as a high chemical shift of the C−5 signal (δ_C_ 74.7 ppm). Residue **B** (δ_H_/δ_C_ 5.20/101.0 ppm, ^1^*J*_C-1,H-1_ ~176 Hz) was recognized as the 3,4-disubstituted L-*glycero*-α-D-*manno*-Hep*p* residue on the basis of the small vicinal couplings between H−1, H−2 and H−3 and the relatively high chemical shifts of the C−3 (δ_C_ 75.3 ppm) and the C−4 (δ_C_ 74.5 ppm) signals. Residue **C** (δ_H_/δ_C_ 4.48/103.5 ppm, ^1^*J*_C-1,H-1_ ~162 Hz) was recognized as the β-D-Glc*p* on the basis of the large vicinal couplings between all protons in the sugar ring. Residue **D** (δ_H_/δ_C_ 5.42/99.7 ppm, ^1^*J*_C-1,H-1_ ~176 Hz) was recognized as the 3,7-disubstituted L-*glycero*-α-D-*manno*-Hep*p* residue from the ^1^H and ^13^C chemical shift values, small vicinal couplings between H−1, H−2 and H−3, and relatively high chemical shift values of the C−3 (δ_C_ 80.2 ppm), and C−7 (δ_C_ 73.1 ppm) signals. Residue **E** (δ_H_/δ_C_ 4.96/102.8 ppm, ^1^*J*_C-1,H-1_ ~172 Hz) was recognized as the 7-substituted L-*glycero*-α-D-*manno*-Hep*p* from the ^1^H and ^13^C chemical shifts, the small vicinal couplings between H−1, H−2 and H−3, and the relatively high chemical shift value of the C−7 (δ_C_ 72.2 ppm) signal. Residue **F** (δ_H_/δ_C_ 5.23/99.7 ppm, ^1^*J*_C-1,H-1_ ~176 Hz) was recognized as the 4-substituted α-D-Gal*p*A based on the characteristic five proton spin system, the high chemical shifts of the H-3 (δ_H_ 4.23 ppm), H-4 (δ_H_ 4.59), H-5 (δ_H_ 4.47) and C-4 (δ_C_ 77.4 ppm) signals, the large vicinal couplings between H-2 and H-3 and small vicinal coupling between H−3, H−4 and H−5. Residue **I** (δ_H_/δ_C_ 5.50/102.0 ppm, ^1^*J*_C-1,H-1_ ~176 Hz) was also recognized as the 4-substituted α-D-Gal*p*A residue based on the similar characteristic five proton spin system. Residue **G** (δ_H_/δ_C_ 5.33/96.1 ppm, ^1^*J*_C-1,H-1_ ~176 Hz) was recognized as the terminal α-D-Glc*p*N due to the large coupling between H−1, H−2 and H−3 and the small vicinal coupling between H−3, H−4 and H−5, as well as the chemical shift value of the C−2 (δ_C_ 55.0). Residue **J** (δ_H_/δ_C_ 5.29/96.5 ppm, ^1^*J*_C-1,H-1_ ~176 Hz) was recognized as the terminal α-D-Glc*p*N due to the large coupling between H−1, H−2 and H−3 and the small vicinal coupling between H−3, H−4 and H−5, as well as the characteristic chemical shift value of the C−2 (δ_C_ 55.7 ppm). Residue **K** (δ_H_/δ_C_ 5.11/99.6 ppm, ^1^*J*_C-1,H-1_ ~174 Hz) was recognized as the 4-substituted α-D-Glc*p*NAc from a low ^13^C chemical shift of the C-2 signal (δ_C_ 52.5 ppm), and of the C−4 signal (δ_C_ 75.7 ppm), and the large vicinal couplings between all ring protons. The N-acetyl group at δ_H_/δ_C_ 2.09/22.7 ppm (δ_C_ 175.8 ppm) was identified. The terminal residue **L** (δ_H_/δ_C_ 5.06/99.8 ppm, ^1^*J*_C-1,H-1_ ~172 Hz) was recognized as the α-D-Glc*p*NAc from a low ^13^C chemical shift of the C−2 signal (δ_C_ 51.6 ppm), and the large vicinal couplings between all ring protons. The N-acetyl group at δ_H_/δ_C_ 2.13/22.7 ppm (δ_C_ 175.8 ppm) was identified. The presence of heterogeneity in OSVI was due to partial replacement of α-D-Glc*p*N (residue **J**) by α-D-Glc*p*NAc (residue **K**). The ^31^P NMR spectra showed no indication of phosphate groups in the OSVI.

In the HSQC-DEPT spectra of OSVI (at δ_H_/δ_C_ 3.92, 4.07/41.8 ppm), additional negative CH_2_ signals were detected. These resonances showed a correlation with carbonyl carbon signals at δ_C_ 168.1 ppm in the HMBC spectra, suggesting the presence of glycine.

The ^1^H−^1^H NOESY spectrum showed strong inter-residue cross-peaks between the following transglycosidic protons: H−1 of **B**/H−5 of **A**, H−1 of **D**/H−3 of **B**, H−1 of **E**/H−7 of **D**, H−1 of **F**/H−7 of **E**, H−1 of **G**/H−4 of **F**, H−1 of **I**/H−3 of **D**, H−1 of **J**/H−4 of **I**, H−1 of **K**/H−4 **I**, H−1 of **L**/H−4 of **K**. The linkage between H−1 of **C**/H−4 of **B** was not observed. Despite this the HMBC spectrum of OSVI confirmed the substitution positions of all monosaccharide residues. The cross-peaks between H−1 of **K**/H−4 of **I**, and H−1 of **J**/H−4 of **I** showed heterogeneity due to the presence of a core-oligosaccharide structure with terminal α-D-Glc*p*N (residue **J**) instead α-D-Glc*p*NAc-(1→4)-α-D-Glc*p*NAc-(1→) fragment (residues **L** and **K**).

#### 2.2.3. Structural Analysis of *E. ictaluri* AL-15-01-CATFISH Core Oligosaccharide

The analysis of *E. ictaluri* OSVII showed the presence of →3,4)-L-*glycero*-α-D-*manno*–Hep*p* (residue **B**), terminal β-D-Glc*p* (residue **C**), →3,7)-L-*glycero*-α-D-*manno*-Hep*p* (residue **D**), L-*glycero*-α-D-*manno*-Hep*p* (residue **E**), →4-α-D-Gal*p*A (residues **I**), →3-α-D-Glc*p*NAc (residue **K**), terminal β-D-Gal*p* (residue **M**) and →5-substituted Kdo*p* (residue **A**) (Figure 8, Table 3).

Residue **A** was identified as the 5-substituted Kdo on the basis of characteristic deoxy proton signals at δ_H_ 1.92/2.27 ppm (H-3*ax*/H-3*eq*) as well as a high chemical shift of the C−5 signal (δ_C_ 74.7 ppm). Residue **B** (δ_H_/δ_C_ 5.06/102.2 ppm, ^1^*J*_C-1,H-1_ ~176 Hz) was recognized as the 3,4-disubstituted L-*glycero*-α-D-*manno*-Hep*p* residue on the basis of the small vicinal couplings between H−1, H−2 and H−3, and relatively high chemical shifts of the C−3 (δ_C_ 75.2 ppm) and the C−4 (δ_C_ 76.1 ppm) signals. Residue **C** (δ_H_/δ_C_ 4.48/103.3 ppm, ^1^*J*_C-1,H-1_ ~162 Hz) was recognized as the β-D-Glc*p* on the basis of the large vicinal couplings between all protons in the sugar ring. Residue **D** (δ_H_/δ_C_ 5.36/99.1 ppm, ^1^*J*_C-1,H-1_ ~176 Hz) was recognized as the 3,7-disubstituted L-*glycero*-α-D-*manno*-Hep*p* residue from the ^1^H and ^13^C chemical shift values, small vicinal couplings between H−1, H−2 and H−3, and relatively high chemical shift values of the C−3 (δ_C_ 76.3 ppm), and C−7 (δ_C_ 70.8 ppm) signals. Residue **E** (δ_H_/δ_C_ 4.96/101.5 ppm, ^1^*J*_C-1,H-1_ ~172 Hz) was recognized as the L-*glycero*-α-D-*manno*-Hep*p* from the ^1^H and ^13^C chemical shifts, the small vicinal couplings between H−1, H−2 and H−3, and the relatively low chemical shift value of the C−7 (δ_C_ 63.9 ppm) signal. Residue **K** (δ_H_/δ_C_ 5.00/98.4 ppm, ^1^*J*_C-1,H-1_ ~172 Hz) was recognized as the 3-substituted α-D-Glc*p*NAc from a low ^13^C chemical shift of the C−2 signal (δ_C_ 54.3 ppm), C−3 (δ_C_ 79.2 ppm) and the large vicinal couplings between all ring protons. The N-acetyl group at δ_H_/δ_C_ 2.19/22.6 ppm (δ_C_ 175.2 ppm) was identified. The residue **M** (δ_H_/δ_C_ 4.54/103.7 ppm, ^1^*J*_C-1,H-1_ ~162 Hz) was recognized as the terminal β-D-Gal*p* due to the large vicinal couplings between H−1, H−2 and H−3 and the small vicinal couplings between H−3, H−4 and H−5. The ^31^P NMR spectra showed no indication of phosphate groups in the OSVI.

In the ^1^H-^13^C HSQC-DEPT spectrum of *E. ictaluri* OSVII (at δ_H_/δ_C_ 3.90, 4.09/41.6 ppm) additional negative CH_2_ signals were detected. These resonances showed a correlation with carbonyl carbon signals at δ_C_ 168.2 ppm in the HMBC spectra, suggesting the presence of glycine.

^1^H−^1^H NOESY spectra showed strong inter-residue cross-peaks between the following transglycosidic protons: H−1 of **D**/H−3 of **B**, H−1 of **E**/H−7 of **D**, H−1 of **I**/H−3 of **D**, H−1 of **K**/H−4 of **I** and H−1 of **M**/H−3 of **K** (Figure 8C).

**Table 3 ijms-24-04768-t003:** Chemical shifts of the *E. ictaluri* AL-15-01-CATFISH core oligosaccharide (OSVII).

Residues	Chemical Shifts (ppm)							
	H1/C1	H2/C2	H3(H3ax,eq)/C3	H4/C4	H5/C5	H6,6’/C6	H7,7’/C7	H8,8’/C8 (NAc)
**A** →5)-Kdo	nd	nd	1.92, 2,27 34.1	4.14 64.6	4.14 74.7	4.01 71.1	3.92 66.9	3.69, 3.89 63.8
**B** →3,4)-L-*glycero*-α-D-*manno*-Hep*p*-(1→	5.18 101.2	4.15 70.4	4.25 75.2	4.28 76.1	4.11 71.0	4.11 69.5	3.79 63.8	
**C** β-D-Glc*p*-(1→	4.48 103.3	3.33 73.4	3.52 75.8	3.52 69.7	3.40 76.9	3.75, 3.88 61.2		
**D** →3,7)-L-*glycero*-α-D-*manno*-Hep*p*-(1→	5.36 99.1	4.26 68.4	4.18 76.3	4.11 67.4	3.79 71.0	4.05 69.5	3.67, 3.89 70.8	
**E** L-*glycero*-α-D-*manno*-Hep*p*-(1→	4.96 101.5	4.00 70.7	3.96 72.7	3.97 69.3	3.87 71.1	4.13 69.3	3.72, 3.92 63.9	
**I** →4)-α-D-Gal*p*A-(1→	5.57 100.8	3.93 69.7	4.18 68.7	4.11 77.9	4.76 71.3	175.1		
**K** →3)-α-D-Glc*p*NAc-(1→ **M** β-D-Gal*p*-(1→	5.00 98.4 4.54 103.7	4.06 54.3 3.62 71.5	3.84 79.2 3.72 73.3	4.01 71.1 3.97 69.2	4.30 71.3 3.67 75.9	3.82, 3.96 60.4 3.70, 3.73 61.1		(2.19) (22.6, 175.2)
Gly	168.4	3.90, 4.09 41.6						

ax, axial position; eq, equatorial position; nd, not detected.

## 3. Disscussion

The *Edwardsiella* genus belongs to the order *Enterobacteriales* and family *Hafniaceae*. At present, this genus includes five species: *E. tarda, E. piscicida*, *E. anguillarum*, *E. hoshinae* and *E. ictaluri.* However, the description of *E. piscicida* [20] and *E. anguillarum* [21] resulted from a reclassification of diverse isolates previously identified as *E. tarda*. Until now, only the chemical analysis and genomics of the core oligosaccharide from *E. tarda* EIB202 have been described [19]. In this report, the chemical structure of the complete LPS-core of *E. piscicida*, *E. anguillarum*, *E. hoshinae* and *E. ictaluri* and the genomic regions (*waa* cluster) involved in its biosynthesis were presented. We identified orthologous genes to *E. tarda* EIB202 *coaD* and *hldD* in seven *E. tarda*, thirteen *E. piscicida*, two *E. anguillarum*, six *E. ictaluri* and two *E. hoshinae* strains with complete genome sequences available on the NCBI website to locate the *waa* gene cluster in each genome. As expected, the genes and genomic organization of this cluster were highly conserved in all species, encoding all the activities required for outer core assembly and the transferases needed for inner core oligosaccharide synthesis. However, comparative genomics of this region by Mauve software and a phylogenetic tree showed three different types of *waa* clusters in *Edwardsiella*. Two types contained a gene that encodes a glycosyltransferase orthologous to WabK and were found in all *E. piscicida*, *E. anguillarum* and *E. ictaluri* strains tested, as well as in three *E. tarda* strains. In *E. tarda* EIB202, WabK was described as a galactosyltransferase that incorporates a Gal*p* residue in a β(1→4) to Glc*p*NAc. However, the *E. ictaluri waa* cluster had a deletion between *wabK* and *wabH*, so the cluster did not contain either *wapC* or part of *wapB.* The loss of this genomic fragment was detected in all the *E. ictaluri* strains whose genomes had been completely sequenced. The third type contained a gene that encodes a glycosyltransferase orthologous to WahX and was found in all *E. hoshinae* strains tested and in four *E. tarda* strains. Furthermore, the antigen O ligase encoded for the *waaL* included in these three *waa* clusters was not orthologous, and this could be related to the differential glycosyltransferase (WabK or WahX) encoded in each of these clusters.

In the last year, a revision of the taxonomic position of the isolates previously identified as *E. tarda* has been carried out, and some strains were reassigned as *E. piscicida* or *E. anguillarum*. This has been the case for *E. tarda* EIB202, FL6-60 and ET-001, which have been reclassified as *E. piscicida* [5]. The reclassification of these strains was in agreement with the phylogenetic tree generated by the neighbor-joining method on the basis of the *waa* cluster sequence that clusters them with *E. piscicida* strains. However, the remaining *E. tarda* strains genomically analyzed were clustered with *E. hoshinae,* and their *waa* clusters contain the same *wahX* gene as *E. hoshinae* strains.

In *E. tarda* EIB202, the *ETAE_RS09105* gene was located outside the *waa* cluster and encodes the WapG transferase, which was involved in the LPS motif β-D-Glc*p*-(1→2)-α-L-Hep*p*II. Orthologous to this protein only were found in *E. piscicida* or *E. tarda,* reclassified as *E. piscicida,* and in some *E. anguillarum* strains (C-5-1 strain). Comparative genomics showed that this chromosomal region was conserved in *E. piscicida* and *E. anguillarum* strains containing the orthologous *ETAE_RS09105* gene but was highly variable in other *Edwardsiella* species.

The chemical structure of the core oligosaccharides carried out in *E. piscicida* HL9.1 and *E. anguillarum* 205/03 seems to be in agreement with the genomic studies, which determined that both strains have the same LPS-core as *E. tarda* EIB202 [19].

Genomic studies conducted by comparison with the BLASTp algorithm showed that *E. ictaluri* strains, such as *E. piscicida* and *E. anguillarum,* had an orthologous to the WabK but did not contain the genes encoding the WapC, WapB or WapG. WapB is an enzyme that transfers α-D-Glc*p*NAc to α-D-Gal*p*A; WapC transfers α-D-Gal*p*A to a Hep*p;* and WapG transfers β-D-Glc*p* to a Hep*p*, all of them in different acceptor substrates of LPS-core in a α(1→4), α(1→7) and α(1→2) linkage, respectively. The chemical structure of the core oligosaccharide carried out in *E. ictaluri* AL-15-01-CATFISH seems to be in agreement with the absence of these three genes.

Genomic studies conducted by comparison with the BLASTp algorithm showed that *E. hoshinae* strains did not have a gene that encodes a protein orthologous to WabK, in contrast to *E. piscicida*, *E. anguillarum* and *E. ictaluri*. However, they had a gene that encodes a N-acetylhexosamine glycosyltransferase (named WahX) that could be involved in the transfer of α-D-Glc*p*NAc to a α-D-Glc*p*NAc or α-D-GlcN in a different acceptor substrate of LPS-core in a α(1→4) linkage (Figure 6). Furthermore, *E. hoshinae* strains, such as *E. ictaluri*, did not have a gene encoding WapG. The presence of an orthologous to WahX and the absence of WapG were also determined in the *E. tarda* strains KC-Pc-HB1, FL95-01, ATCC15947 and AL98-87. The chemical structure of the core oligosaccharide carried out in *E. hoshinae* DSMZ 13771^T^ seemed to be in agreement with the absence of WabK and the presence of WahX.

Figure 9 shows the presumptive chemical structure and assignment of genes involved in the three *Edwardsiella* core oligosaccharide types. The first type was found in *E. piscicida* and some *E. anguillarum* strains, as well as in *E. tarda* EIB202. In *E. anguillarum* ET080813, no orthologous to WapG was detected, and maybe it does not have the H residue. The second type was found in *E. ictaluri,* and the third type was found in *E. hoshinae* strains.

## 4. Materials and Methods

### 4.1. Bacterial Strains and Culture Conditions

Bacterial strains from the *Edwardsiella* genus used in these experiments are described as follows: *E. tarda* EIB202 was obtained from the Y. Zhang laboratory [22]; *E. piscicida* HL9.1, *E. hoshinae* DSMZ 13771^T^ and *E. anguillarum* 205/03 strains were provided by the B. Magariños group in the University of Santiago de Compostela; and the *E. ictaluri* AL-15-01-CATFISH strain was provided by the C. R. Arias group in Auburn University. *Edwardsiella* strains were grown in nutrient broth (NB). While *E. tarda* and *E. piscicida* were grown at 37 °C, *E. anguillarum*, *E. hoshinae* and *E. ictalurid* were grown at 26 °C.

### 4.2. LPS Isolation and Electrophoresis

For screening purposes, LPS was obtained after proteinase K digestion of whole cells, and the LPS samples were separated by SDS-Tricine-PAGE as previously described and visualized by silver staining [23,24]. Two different methods based on the hot phenol-water method described by Westphal were performed in order to extract the LPS from bacterial cells [25]. The first variation was performed with *E. piscicida* and *E. anguillarum*, which produce an O-antigen with many repeating units. The LPS was extracted from the crude-cell envelope fraction according to the Westphal method, modified by Osborn [25,26].

### 4.3. Core Oligosaccharide Isolation

The LPS (20 mg) was heated with 1.5% acetic acid at 100 °C for 45 min. The precipitate (lipid A) was removed by centrifugation (20 min/12,000 rpm/4 °C). The supernatants, containing the mixture of poly- and oligosaccharides, were fractionated using a Bruker chromatographic system (amaZon SL, Berlin, Germany) on a Superdex G2500 column (7.5 mm × 60 cm) equilibrated with 0.05 M acetic acid and with a flow rate of 1 mL/min each fraction. Eluates were monitored with a Knauer differential refractometer, and all fractions were checked by NMR spectroscopy.

### 4.4. Instrumental Method

NMR spectroscopy. All NMR spectra were recorded on a Bruker Avance III 600 MHz spectrometer equipped with a 5 mm QCI cryoprobe with *z*-gradient. The measurements were performed at 298 K without simple spinning and using the acetone signal (𝛿_H_/𝛿_C_ 2.225/31.05 ppm) as an internal reference. In the TOCSY experiments, the mixing times were 30, 60 and 100 ms. The NOESY experiment was performed with a mixing time of 200 ms, and the HMBC experiment with a delay of 80 ms. The data were acquired and processed using standard Bruker software (TopSpin 3.0). The processed spectra were assigned with the help of the SPARKY program [27].

### 4.5. Comparative Genomics

The complete assembled genome sequences of seven *E. tarda*, thirteen *E. piscicida*, two *E. anguillarum*, six *E. ictaluri* and two *E. hoshinae* strains available on the National Center for Biotechnology Information (NCBI) genome website were retrieved, and we performed a local *tblastn* of *E. tarda* EIB202 *coaD* and *hldD* (also referred to as *rfaD*) to locate the *waa* gene cluster in each genome. We also performed a local *tblastn* of *E. tarda* EIB202 *ETAE_RS09105*. The genomic regions containing these genes were compared through progressive Mauve software (https://darlinglab.org/mauve/mauve.html accessed on 3 November 2022) using the default parameters [28] and genes in the selected region were predicted with Glimmer v3.0.2 [29]. Alignment of genomic regions containing the *waa* cluster was performed by MUSCLE [30] of the EMBL-EBI website [31]. For each strain, the Genome Assembly and Annotation report was of interest in order to see the proteins corresponding to the core oligosaccharide part of the LPS. Protein domains were determined with the NCBI Conserved Domain Database (CDD) [32], and identities were inspected using the BLASTp network service at NCBI. Phylogenetic tree diagrams were generated by the neighbor-joining method using the MEGA-X software version 10.0.4 [33].

## Figures and Tables

**Figure 1 ijms-24-04768-f001:**
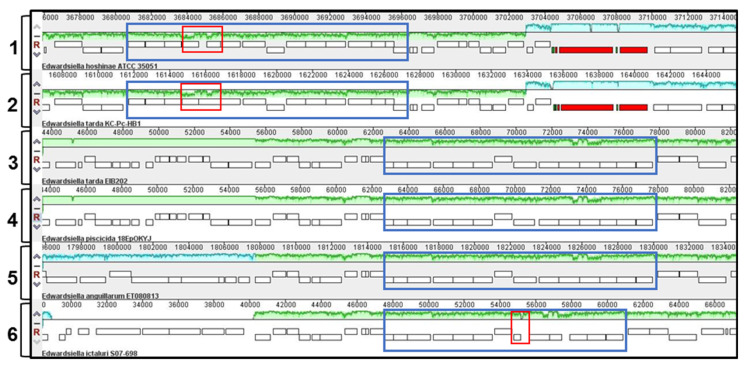
Comparative genomic analysis using progressive Mauve to identify the *waa* cluster on the chromosomes of *E. hoshinae* ATCC35051 (track 1), *E. tarda* KC-Pc-HB1 (track 2), *E. tarda* EIB202 (track 3), *E. piscicida* 18EpOKYJ (track 4), *E. anguillarum* ET080813 (track 5) and *E. ictaluri* S07-698 (track 6). Matching colors indicate homologous segments that are connected across genomes. Chromosomal regions inside blue squares contain *waa* clusters, and red squares show differences in comparison to *E. tarda* EIB202.

**Figure 2 ijms-24-04768-f002:**
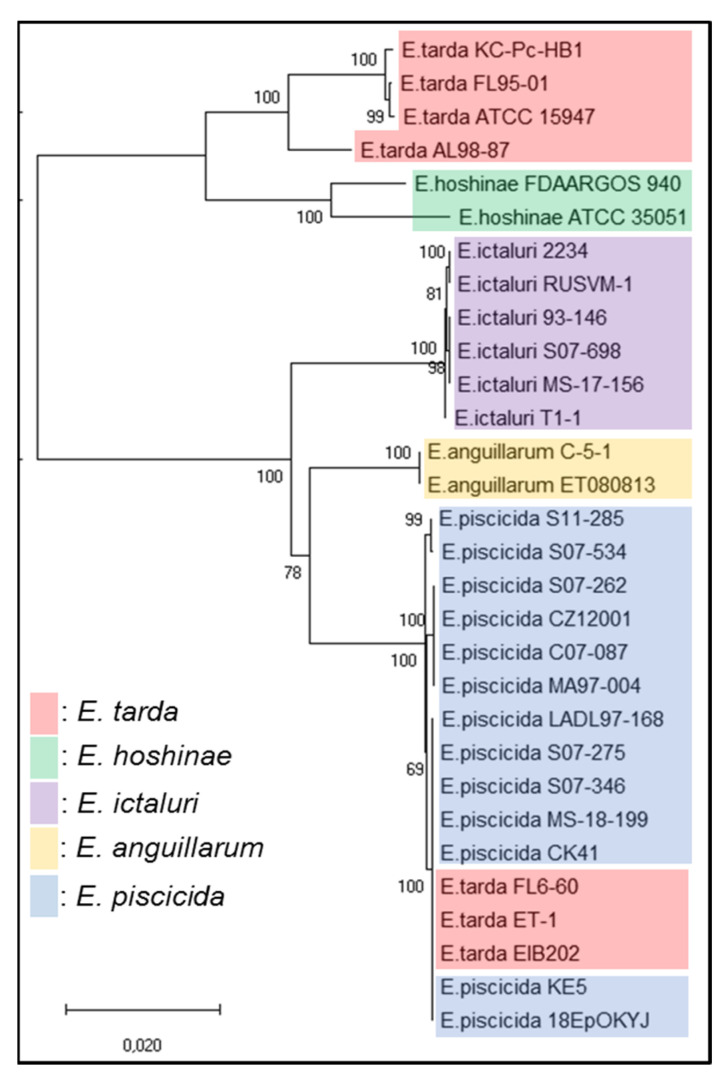
Phylogenetic tree generated by the neighbor-joining method on the basis of the *waa* cluster sequence.

**Figure 3 ijms-24-04768-f003:**
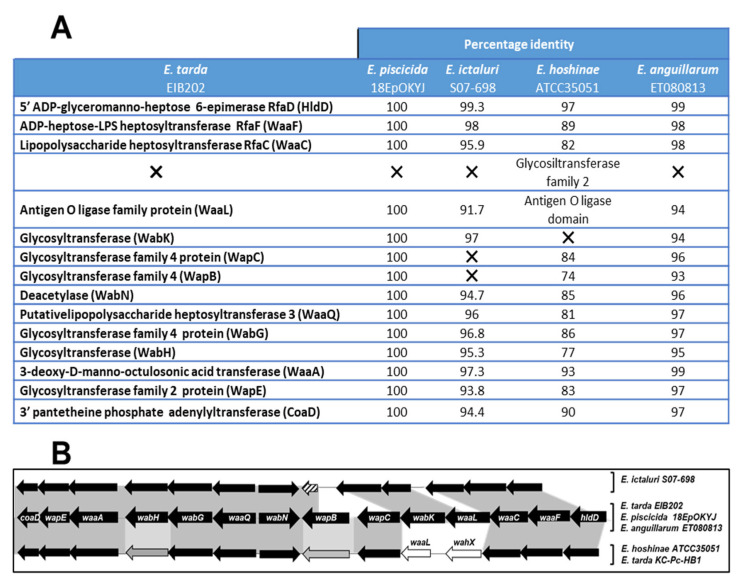
(**A**) Comparison of the proteins present in the *waa* gene cluster of four different *Edwardsiella* species. (**B**) Schematic comparison of *waa* gene cluster models of *Edwardsiella.* Black arrows and dark gray color between sequences indicate identities higher than 80.0%. Gray arrows and a light gray color between sequences indicate identities higher than 70.0%. White arrows and no color between sequences indicate no identities. Striped arrows indicate deleted genes.

**Figure 4 ijms-24-04768-f004:**
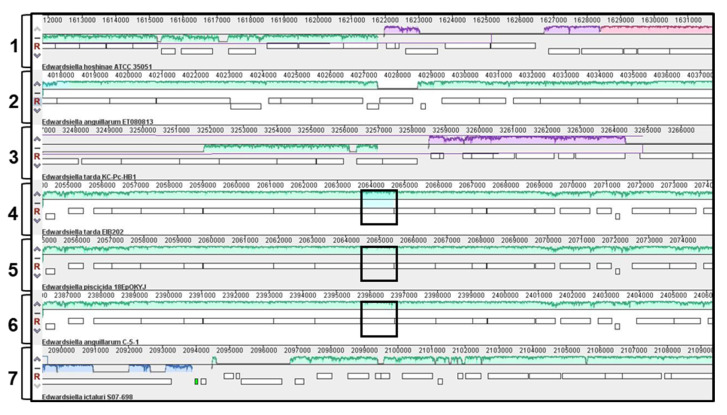
Comparative genomic analysis using progressive Mauve to identify the chromosomal region containing orthologous to *wapG* of *E. tarda* EIB202 in the chromosomes of *E. hoshinae* ATCC35051 (track 1), *E. anguillarum* ET080813 (track 2), *E. tarda* KC-Pc-HB1 (track 3), *E. tarda* EIB202 (track 4), *E. piscicida* 18EpOKYJ (track 5), *E. anguillarum* C-5-1 (track 6) and *E. ictaluri* S07-698 (track 7). Matching colors indicate homologous segments connected across genomes. Chromosomal regions inside black squares contain orthologous to *wapG*.

**Figure 5 ijms-24-04768-f005:**
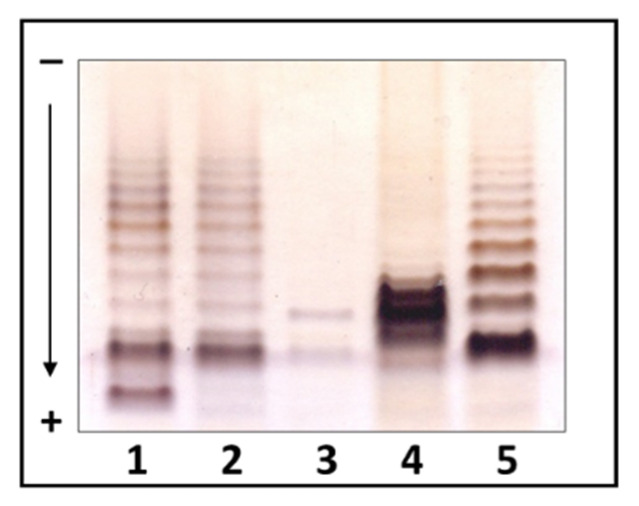
*Edwardsiella* lipopolysaccharides analyzed by tricine SDS-PAGE*. E. tarda* EIB202 (lane 1), *E. piscicida* HL9.1 (lane 2), *E. ictaluri* AL-15-01-CATFISH, (lane 3), *E. hoshinae* DSMZ 13771^T^ (lane 4) and *E. anguillarum* 205/03 (lane 5).

**Figure 6 ijms-24-04768-f006:**
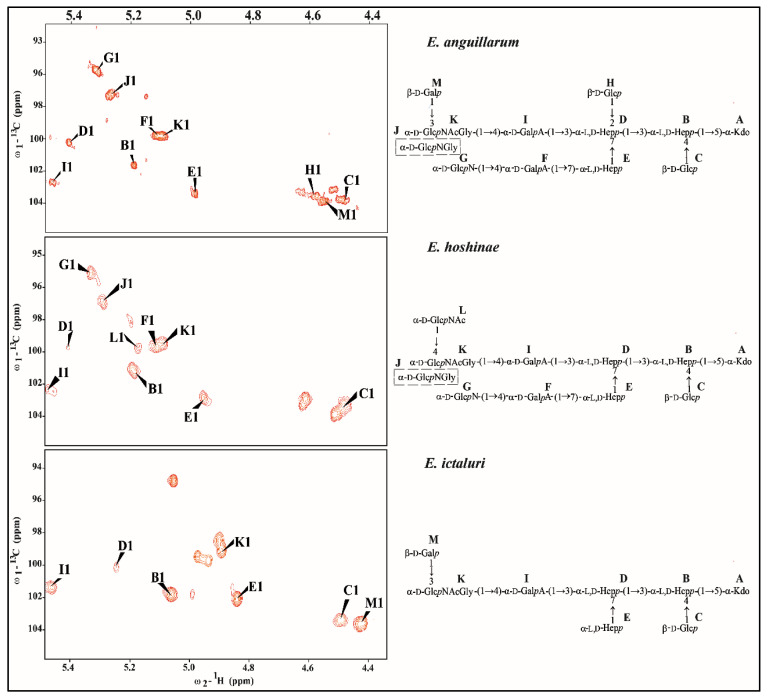
Anomeric regions of ^1^H−^13^C HSQC-DEPT spectra of *E. anguillarum* 205/03*, E. hoshinae* DSMZ 13771^T^ and *E. ictaluri* AL-15-01-CATFISH core oligosaccharides.

**Figure 7 ijms-24-04768-f007:**
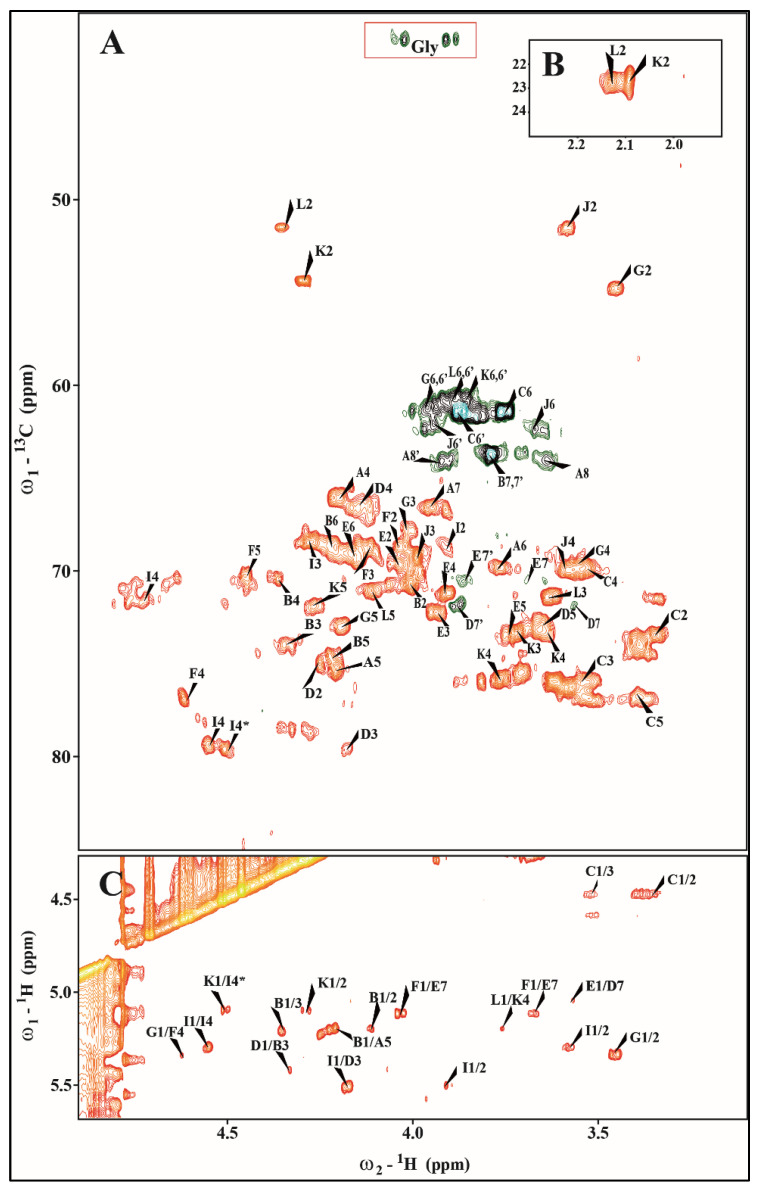
(**A**,**B**) Selected regions of the ^1^H−^13^C HSQC-DEPT and (**C**) ^1^H−^1^H NOESY spectra of the fraction OSVI of *E. hoshinae* DSMZ 13771^T^ core oligosaccharide.

**Figure 8 ijms-24-04768-f008:**
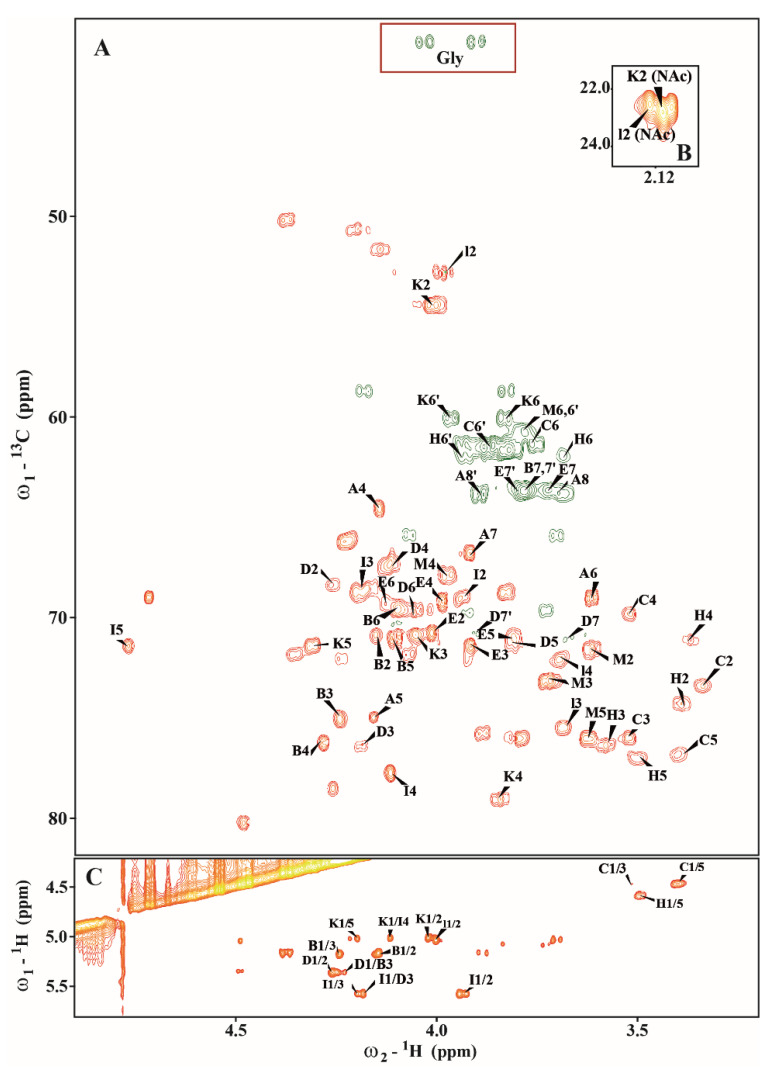
(**A**,**B**) Selected regions of the ^1^H−^13^C HSQC-DEPT and (**C**) ^1^H−^1^H NOESY spectra of the fraction OSVII of *E. ictaluri* AL-15-01-CATFISH core oligosaccharide.

**Figure 9 ijms-24-04768-f009:**
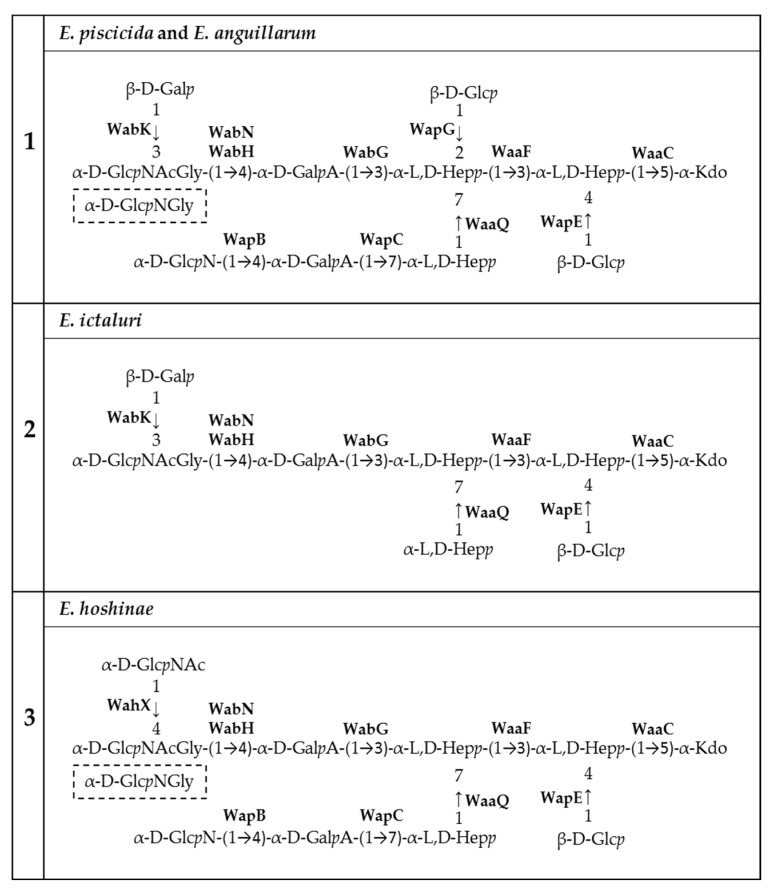
(**1**) Chemical structure of the core oligosaccharide of *E. piscicida* HL9.1 and *E*. *anguillarum* 205/03, (**2**) *E. ictaluri* AL-15-01-CATFISH, and (**3**) *E. hoshinae* DSMZ 13771^T^ with presumptive assignment of genes involved in its biosynthesis.

**Table 1 ijms-24-04768-t001:** Chemical shifts of the *E. anguillarum* 205/03 core oligosaccharide (OSVII).

Residues	Chemical Shifts (ppm)							
	H1/C1	H2/C2	H3(H3ax,eq)/C3	H4/C4	H5/C5	H6,6’/C6	H7,7’/C7	H8,8’/C8 (NAc)
**A** →5)-Kdo	nd	97.6	1.95, 2,32 34.6	4.19 66.3	4.25 75.3	3.77 69.8	3.92 66.7	3.70, 3.89 64.4
**B** →3,4)-L-*glycero*-α-D-*manno*-Hep*p*-(1→	5.17 101.3	4.14 69.8	4.32 75.0	4.31 74.2	4.25 71.9	4.15 70.2	3.82 63.7	
**C** β-D-Glc*p*-(1→	4.55 103.3	3.38 74.4	3.51 76.3	3.49 69.9	3.45 76.5	3.86, 3.89 62.0		
**D** →2,3,7)-L-*glycero*-α-D-*manno*-Hep*p*-(1→	5.44 99.6	4.34 78.5	4.11 80.0	4.11 66.6	3.69 73.3	4.28 69.3	3.63, 4.01 73.3	
**E** →7)−L-*glycero*-α-D-*manno*-Hep*p*-(1→	4.97 103.4	4.05 71.1	3.95 72.3	3.93 71.4	3.72 73.4	4.23 69.5	3.69, 3.88 72.0	
**F** →4)-α-D-Gal*p*A-(1→	5.42 99.4	4.09 70.0	4.20 68.5	4.62 77.6	4.44 70.4	176.5		
**G** α-D-Glc*p*N-(1→	5.27 95.5	3.36 55.4	4.00 70.4	3.65 68.3	4.14 73.3	3.68, 3.95 60.4		
**H** β-D-Glc*p*-(1→	4.63 103.1	3.40 74.8	3.56 71.3	3.33 71.3	3.62 76.4	3.67, 3.92 62.0		
**I** →4)-α-D-Gal*p*A-(1→	5.42 102.5	3.92 69.9	4.29 72.2	4.48 80.9	4.63 72.4	175.5		
**J** α-D-Glc*p*N-(1→	5.29 97.2	3.39 55.6	3.98 70.6	3.63 70.5	4.28 72.5	3.70, 3.98 62.5		
**K** →3)-α-D-Glc*p*NAc-(1→ **M** β-D-Gal*p*-(1→	5.20 99.6 4.53 103.7	4.27 51.5 3.61 71.3	3.78 75.4 3.79 72.2	3.64 71.1 3.90 71.2	4.03 71.3 3.59 75.3	3.82, 3.85 60.1 3.73, 3.77 63.1		2.13 22.7, 175.7
Gly	168.2	3.92, 4.07 41.8						

ax, axial position; eq, equatorial position. nd, not detected.

**Table 2 ijms-24-04768-t002:** Chemical shifts of the *E. hoshinae* DSMZ 13771^T^ core oligosaccharide (OSVI).

Residues	Chemical Shifts (ppm)							
	H1/C1	H2/C2	H3(H3ax,eq)/C3	H4/C4	H5/C5	H6,6’/C6	H7,7’/C7	H8,8’/C8 (NAc)
**A** →5)-Kdo	nd	nd	1.93, 2,26 34.2	4.12 66.0	4.21 74.7	3.73 69.8	3.90 66.4	3.74, 3.85 64.3
**B** →3,4)-L-*glycero*-α-D-*manno*-Hep*p*-(1→	5.20 101.0	4.15 69.8	4.29 75.3	4.32 74.5	4.20 71.7	4.17 70.3	3.81 63.4	
**C** β-D-Glc*p*-(1→	4.48 103.5	3.38 74.7	3.53 76.0	3.39 70.1	3.29 75.9	3.76, 3.90 62.2		
**D** →3,7)-L-*glycero*-α-D-*manno*-Hep*p*-(1→	5.42 99.7	4.32 74.8	4.13 80.2	4.12 66.9	3.71 73.3	4.24 69.3	3.64, 4.01 73.1	
**E** →7)−L-*glycero*-α-D-*manno*-Hep*p*-(1→	4.96 102.8	4.06 71.3	3.91 72.2	3.97 71.2	3.79 73.1	4.25 69.1	3.71, 3.80 72.2	
**F** →4)-α-D-Gal*p*A-(1→	5.23 99.7	4.13 70.1	4.23 68.2	4.59 77.4	4.47 70.3	176.2		
**G** α-D-Glc*p*N-(1→	5.33 96.1	3.34 55.0	4.04 70.0	3.69 68.1	4.11 73.2	3.71, 3.98 60.3		
**I** →4)-α-D-Gal*p*A-(1→	5.50 102.0	3.93 70.0	4.25 72.1	4.49 80.2	4.63 72.6	175.8		
**J** α-D-Glc*p*N-(1→	5.29 96.5	3.42 55.7	3.86 70.3	3.68 70.2	4.20 72.3	3.72, 3.99 62.4		
**K** →4)-α-D-Glc*p*NAc-(1→	5.11 99.6	4.29 52.5	3.64 73.3	3.58 74.6	4.261 71.9	3.91,3.94 60.8		2.09 22.7, 175.8
**L** α-D-Glc*p*NAc-(1→	5.21 99.7	4.36 51.6	3.62 71.5	3.64 73.4	4.11 71.1	3.85, 3.87 60.7		2.13 22.7, 175.8
Gly	168.2	3.92, 4.07 41.8						

ax, axial position; eq, equatorial position. nd, not detected.

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
