# Peer review of "Structural Diversity among Edwardsiellaceae Core Oligosaccharides"

_ijms, 2023, doi:10.3390/ijms24054768_

Round 1

Reviewer 1 Report

The manuscript subjected to IJMS, entitled „Structural diversity among Edwardsiellaceae core oligosaccharides” by María Jordán Ramos, Sylwia Wojtys-Tekiel, Susana Merino , Juan M. Tomás and Marta Kaszowska, concerns chemical structure of core region of LPS isolated from five different pathogenic species of bacteria belonging to the Edwardsiella genus and the genomic context of their biosynthesis. The structural studies were well conducted, using only 2D NMR spectroscopy and SDS-PAGE techniques. Anyway, the mass spectrometry methods (e.g. MALDI-TOF MS-MS spectra) could be very useful to establish the core region structural diversity among the group of investigated strains, and showing three models of Edwardsiella core region. The comparative genomic studies on the genes involved in the core region biosynthesis were also detaily described.

The Authors did not inform in the Introduction section, that the detailed structure of the core region of one of the strains used in current studies has been already published (Kaszowska, M., de Mendoza-Barberá, E., Maciejewska, A., Merino, S., Lugowski, C., J. M. Tomás “The complete structure of the core oligosaccharide from Edwardsiella tarda EIB 202 lipopolysaccharide.” International Journal of Molecular Sciences, 2017, 18 (6), 1163). In this previous paper the structure of core region of Edwardsiella tarda EIB 202 and its biosynthesis pathway were also described in detail. Structural studies were based on the chemical analyses, 2D NMR spectroscopy and MALDI-TOF MS-MS spectrometry techniques. In the current paper Authors present in the Introduction section only the genetic studies on the core region biosynthesis of the strain E. tarda EJB202 and there is no information about the structure, which is already known. Of course, the reviewer understand, that this is continuation of previous work, using five different  species, in the light of strong changes in the taxonomy inside Edwardsiella genus. Anyway, the Introduction should be modified, and some information above previous work of the Authors should be included. Also, the article would be more valuable if Authors decided to include results of chemical analyses and mass spectra (MS-MS) for 3 types of core regions currently described.

Other remarks and suggestions for Authors:

1.       Abstract –  Authors stated that: “The structure of core oligosaccharides was investigated by ¹H and 13C nuclear magnetic resonance (NMR) spectroscopy, matrix-assisted laser-desorption/ionization time-of-flight (MALDI-TOF) mass spectrometry and chemical analysis.” In the article there is nothing about mass spectrometry, nor from any chemical analyses of investigated preparations. Please comment on it or remove this sentence from the abstract.

2.       Abstract - The figure should be moved to the end of Abstract .

3.       Introduction - There is a very scanty information about LPS structure in general. Please add some information, to clarify the subject of interests.

4.       Introduction – Page 2, lines 26-35 – I suggests to move this paragraph to the Discussion section.

5.       Whole manuscript  - there is ”wa gene cluster”, please change to “waa gene cluster”.

6.       Figure 1 – the figure is completely illegible in manuscript form, especially A and B parts. Please improve it or change the construction of tables.

7.       Figure 3 and its description in paragraph 2.2. – Authors stated that “all the LPSs are smooth (S-LPS)”. Line 3 – the material is rather rough or SR (only two strong bands are visible). The OPS of preparation 4 is also rather short.

8.       Figure 4 – the structural formula included in the right part of the figure is illegible (very small fonts were used). Please modify the structural formula.

9.       Table 1  - there is a mistake in chemical shift values assigned to residue H  - glucose residue possess only 6 carbons, not 7, and please check carefully all values for proton/carbon chemical shifts.

10.   Materials and Methods chapter -  Authors should start from Bacterial strains and culture conditions. The genomic studies should be included at the end of this chapter. Additionally, there is no information of any chemical analyses used in this study, which were announced in Abstract. Usually, it should be made total sugar analysis, linkage analysis (methylation), as well as the absolute configuration of sugars should be also performed. Also, there is no details of SDS-PAGE technique used in this study. Please comment of it.

11.   Materials and Methods – there is a lack of description of bacterial culture conditions. Please complete it.

12.   The same chapter, paragraph 4.3. LPS isolation. Two citations (27 and 25) are incorrect.

13.   The same chapter, paragraph 4.4. Core oligosaccharide isolation. There is no information about the amount of LPS used. The sentence: “The precipitate was removed by centrifugation for 20 min. at 12.266 rpm at 4 °C.” contains some inaccuracies, and should be modified, e.g. to the form: “The lipid A precipitate was removed by centrifugation…” The rpm values usually are given with less accuracy (could be: 12 000 rpm). And later: “The mixture of poly- and oligosaccharides were fractionated on Superdex G2500 column (amaZon SL, Germany)” I understand, that Authors used SEC technique to separate poly- and oligosaccharidic fractions, and they used column filled out with Superdex G2500. Here, in brackets, they should write the column size. Afterwards, the information about solvent used and its flow rate should be given. Also, the detector details (RI detector?) should be included. The information given by Authors in brackets: “(amaZon SL, Germany)”, possibly concerns Bruker’s mass spectrometer. Please verify.

14.   References: position 2. is incomplete. Lack of the article title.

Author Response

Please see the attachment which contain the answers to the reviewer 1

Reviewer 2 Report

The submitted manuscript on Structural diversity among Edwardsiellaceae core oligosaccharides present interesting structural information and the presented experimental evidence is of good quality. However the manuscript is too descriptive, lacks a discussion on structure-function relationship and the overall presentation is not sufficiently clear in the present state to guarantee publication in ijms.

In particular the figures should seriously upgraded.

Figure 1 : The quality of the figure is not sufficient to be able to read something. Part C is important, but the quality should be increased. Parts A and B could be placed as supplementary data.

Figure 2 : OK

Figure 3 : the direction of migration was not indicated (- +)

Concerning figure 3, your statement is: “The analyses showed that all LPSs are smooth (S-LPS)”. Could you please explain why ? Any reference ?

You talk about fractions after mild hydrolysis: “The fractions consisting of unsubstituted core oligosaccharide (OS) were identified”. How did you purify those fractions ? In Mat & Meth you talk about gel chromatography (section 4.4 Superdex G2500) but you give no proof.

Figure 4, 5, 6 : the quality should be improved

Author Response

Please see the attachment which contain the answers to the reviewer 2

Round 2

Reviewer 1 Report

The manuscript subjected to IJMS, entitled „Structural diversity among Edwardsiellaceae core oligosaccharides” by María Jordán Ramos, Sylwia Wojtys-Tekiel, Susana Merino , Juan M. Tomás and Marta Kaszowska,  has been already corrected. All my comments and remarks were taken into account by Authors and the manuscript was videly improved.
Thus, I can state, that the presented paper can be accepted without any further changes.

Reviewer 2 Report

The authors accepted and  used almost all of the suggestions. The manuscript is now acceptable